# Electric field causes volumetric changes in the human brain

**Miklos Argyelan[1,2,3]\*, Leif Oltedal[4,5], Zhi-De Deng[6], Benjamin Wade[7], Marom Bikson[8], Andrea Joanlanne[1], Sohag Sanghani[1], Hauke Bartsch[5,9], Marta Cano[10,11], Anders M Dale[9,12,13], Udo Dannlowski[14], Annemiek Dols[15], Verena Enneking[14], Randall Espinoza[16,17], Ute Kessler[4,18], Katherine L Narr[16,17], Ketil J Oedegaard[4,18], Mardien L Oudega[15], Ronny Redlich[14], Max L Stek[15], Akihiro Takamiya[19,20], Louise Emsell[21], Filip Bouckaert[21,22], Pascal Sienaert[22], Jesus Pujol[11,23], Indira Tendolkar[24,25,26], Philip van Eijndhoven[24,25], Georgios Petrides[1,2,3], Anil K Malhotra[1,2,3], Christopher Abbott[27]**

[1]Department of Psychiatry, The Zucker Hillside Hospital, Glen Oaks, United States; [2]Center for Neuroscience, Feinstein Institute for Medical Research, Manhasset, United States; [3]Department of Psychiatry, Zucker School of Medicine, Hempstead, United States; [4]Department of Clinical Medicine, University of Bergen, Bergen, Norway; [5]Department of Radiology, Haukeland University Hospital, Mohn Medical Imaging and Visualization Centre, Bergen, Norway; [6]Experimental Therapeutics and Pathophysiology Branch, National Institute of Mental Health, Bethesda, United States; [7]Department of Neurology, Ahmanson-Lovelace Brain Mapping Center, University of California, Los Angeles, Los Angeles, United States; [8]Department of Biomedical Engineering, The City College of the City University of New York, New York, United States; [9]Center for Multimodal Imaging and Genetics, University of California, San Diego, San Diego, United States; [10]Department of Psychiatry, Bellvitge University Hospital-IDIBELL, Barcelona, Spain; [11]CIBERSAM, Carlos III Health Institute, Barcelona, Spain; [12]Department of Radiology, University of California, San Diego, San Diego, United States; [13]Department of Neurosciences, University of California, San Diego, San Diego, United States; [14]Department of Psychiatry and Psychotherapy, University of Muenster, Muenster, Germany; [15]Department of Psychiatry, Amsterdam UMC, location VUmc, GGZinGeest, Old Age Psychiatry, Amsterdam Neuroscience, Amsterdam, Netherlands; [16]Department of Neurology, University of California, Los Angeles, Los Angeles, United States; [17]Department of Psychiatry and Biobehavioral Sciences, University of California, Los Angeles, Los Angeles, United States; [18]Division of Psychiatry, Haukeland University Hospital, University of Bergen, Bergen, Norway; [19]Department of Neuropsychiatry, Keio University School of Medicine, Tokyo, Japan; [20]Center for Psychiatry and Behavioral Science, Komagino Hospital, Tokyo, Japan; [21]Department of Geriatric Psychiatry, University Psychiatric Center, KU Leuven, Leuven, Belgium; [22]Academic center for ECT and Neurostimulation (AcCENT), University Psychiatric Center, KU Leuven, Kortenberg, Belgium; [23]MRI Research Unit, Department of Radiology, Hospital del Mar, Barcelona, Spain; [24]Department of Psychiatry, Radboud University Medical Center, Nijmegen, Netherlands; [25]Donders Institute for Brain Cognition and Behavior, Centre for Cognitive Neuroimaging, Nijmegen, Netherlands; [26]Faculty of Medicine and LVR Clinic for Psychiatry and Psychotherapy, University of Duisburg-Essen, Essen, Germany; [27]Department of Psychiatry, University of New Mexico School of Medicine, Albuquerque, United States

\*For correspondence: argyelan@gmail.com

**Abstract** Recent longitudinal neuroimaging studies in patients with electroconvulsive therapy (ECT) suggest local effects of electric stimulation (lateralized) occur in tandem with global seizure activity (generalized). We used electric field (EF) modeling in 151 ECT treated patients with depression to determine the regional relationships between EF, unbiased longitudinal volume change, and antidepressant response across 85 brain regions. The majority of regional volumes increased significantly, and volumetric changes correlated with regional electric field (t = 3.77, df = 83, r = 0.38, p=0.0003). After controlling for nuisance variables (age, treatment number, and study site), we identified two regions (left amygdala and left hippocampus) with a strong relationship between EF and volume change (FDR corrected p<0.01). However, neither structural volume changes nor electric field was associated with antidepressant response. In summary, we showed that high electrical fields are strongly associated with robust volume changes in a dose-dependent fashion.

## Introduction

Electroconvulsive therapy (ECT) remains the most effective approach for treatment resistant depressive episodes, as well as the most established neuromodulation technique (*UK ECT Review Group, 2003*; *Fink and Taylor, 2007*). Despite intensive research, however, the mechanism of action for ECT remains unknown, but does involve at least two potentially therapeutic components: electric perturbation and/or seizure activity. One common element across various neuromodulation techniques is the application of different intensities of electric field (EF) on the human brain. Understanding how ECT-induced EF interacts with the cortex and subcortical structures can both advance our mechanistic understanding of ECT and enrich our understanding of other neuromodulation approaches such as magnetic seizure therapy (MST), transcranial magnetic stimulation (TMS), transcranial direct current stimulation (tDCS), and deep brain stimulation (DBS).

A recent longitudinal ECT-imaging study from the Global ECT-MRI Collaboration (GEMRIC) (*Oltedal et al., 2018*) assessed hippocampal volume changes in a large cohort of subjects (N = 268) receiving right unilateral (RUL) or bilateral (BL) electrode placements. The results demonstrated that the volume of the hippocampus increased over the course of ECT treatment and correlated with the number of ECT sessions administered during the ECT series. In addition, the subjects receiving RUL electrode placement had a significantly larger volume change ipsilateral to the side of stimulation, consistent with previous ECT-neuroimaging observations (*Abbott et al., 2014*; *Dukart et al., 2014*; *Pirnia et al., 2016*; *Bouckaert et al., 2016*; *Sartorius et al., 2016*; *Cano et al., 2018*). Our most recent study of 331 subjects with longitudinal MRI scanning pre- and post-ECT showed brain volume increases across several subcortical and cortical regions with strong lateralization of the effects if the electrode placement was RUL (*Ousdal et al., 2019*). Contrary to a priori expectations (*Joshi et al., 2016*; *Cano et al., 2017*), increased volume in these key areas did not translate to better clinical outcome. While the association between the number of ECT sessions and volume change and the laterality of the volume changes all implied a dose–response causative relationship, the role of ECT-mediated neuroplasticity and the underlying mechanism for antidepressant response remains elusive. Furthermore, given the naturalistic design of the studies included for mega-analysis (e.g., non-responders had a longer ECT course and were frequently switched to bilateral treatment at varying intervals), both the number of ECT sessions and electrode placement varied depending on the clinical response, further confounding the dose-response association and its interpretation.

Recent research has challenged the notion that a primary purpose of electric stimulation in treating depression is to generate widespread seizure activity (*Sackeim, 2015*; *Regenold et al., 2015*). Alternatively, electric stimulation may be a therapeutic component of ECT and similar to other non-convulsive neuromodulation treatments. Finite-element simulation was developed to estimate the spatial distribution of the electric field on a voxel-wise basis (*Lee et al., 2012*; *Bikson et al., 2012*). The technique was recently validated in humans (*Huang et al., 2017*). Preliminary computational analyses based on three realistic head models suggested that the ECT electric field distribution had a direct association with clinical and cognitive outcomes, explaining the rationale behind different

**eLife digest** Electroconvulsive therapy, or ECT for short, can be an effective treatment for severe depression. Many patients who do not respond to medication find that their symptoms improve after ECT. During an ECT session, the patient is placed under general anesthesia and two electrodes are attached to the scalp to produce an electric field that generates currents within the brain. These currents activate neurons and make them fire, causing a seizure, but it remains unclear how this reduces symptoms of depression.

For many years, researchers thought that the induced seizure must be key to the beneficial effects of ECT, but recent studies have cast doubt on this idea. They show that increasing the strength of the electric field alters the clinical effects of ECT, without affecting the seizure. This suggests that the benefits of ECT depend on the electric field itself.

Argyelan et al. now show that electric fields affect the brain by making a part of the brain known as the gray matter expand. In a large multinational study, 151 patients with severe depression underwent brain scans before and after a course of ECT. The scans revealed that the gray matter of the patients' brains expanded during the treatment. The patients who experienced the strongest electric fields showed the largest increase in brain volume, and individual brain areas expanded if the electric field within them exceeded a certain threshold. This effect was particularly striking in two areas, the hippocampus and the amygdala. Both of these areas are critical for mood and memory.

Further studies are needed to determine why the brain expands after ECT, and how long the effect lasts. Another puzzle is why the improvements in depression that the patients reported after their treatment did not correlate with changes in brain volume. Disentangling the relationships between ECT, brain volume and depression will ultimately help develop more robust treatments for this disabling condition.

electrode placement strategies in ECT treatment (*Bai et al., 2017*). This finding is in agreement with our previous observation where RUL treatment induced higher volumetric changes in the right hippocampus compared to the left (*Oltedal et al., 2018*), implying that more lateralized electric stimulation rather than a global generalized seizure, may be responsible at least for part of the antidepressant effects of ECT. However, to date, no study has demonstrated the relationship between ECT electric field distribution and treatment response. In this study, we used the large Global ECT-MRI Research Collaboration (GEMRIC) ECT-imaging data set to explicitly determine the relationships between regional 1) electric field strength and volume changes, 2) volume changes and antidepressant response, and 3) electric field and antidepressant response. For the purpose of our primary research question and in contrast to previous GEMRIC investigations, we limited the analyses to subjects that only received right unilateral electrode placement.

## Results

### Clinical results

Subjects showed an average MADRS improvement of 61.3%±33.9% following ECT (pre-ECT MADRS 33.9 (range: 14–54), post-ECT MADRS 12.9 (range: 0–51). Highly significant correlations between age and clinical response (t = 5.75, df = 149, r = 0.43, $p<10^{-7}$, older patients responded better), as well as age and total brain volume (t = −7.32, df = 149, r = −0.51, $p<10^{-10}$) were also observed.

### Laterality of electric field and volume change

ECT was associated with increased volume across all brain regions except the brain stem and bilateral cerebellum cortex (*Supplementary file 1*). In the majority of the regions, right hemisphere regions had greater volumetric change with respect to the corresponding left hemisphere region; no left hemisphere regions had greater volumetric changes compared to the corresponding right-sided region (*Supplementary file 2*, *Figure 1*). Average EF strongly correlated with ΔVol across the ROIs (*Figure 1*, t = 3.77, df = 83, r = 0.38, p=0.0003). To show that this correlation was not simply due to a general effect of the hemisphere (right side had higher EF and volume change while left side had lower values), we calculated laterality indices in both EF and volume change. The correlation

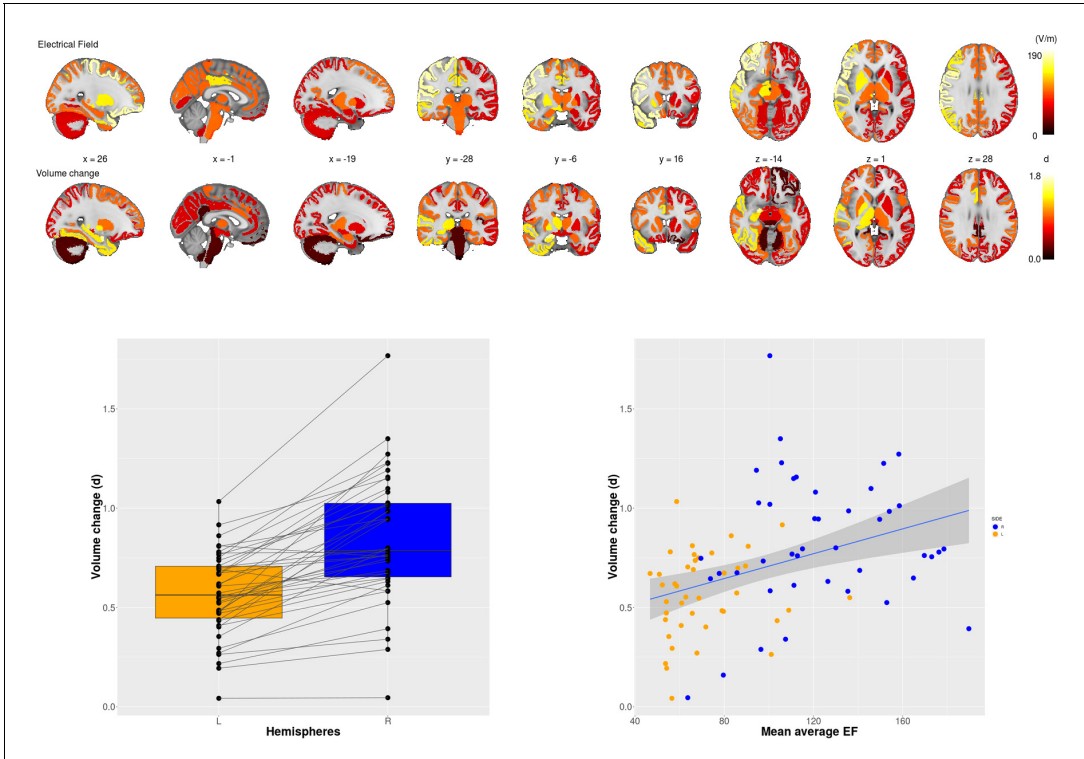

**Figure 1.** Electric Field (EF) and volume change across 85 brain regions. Upper panel first row: Mean EF across 85 brain regions; second row: the effect size of volume changes between baseline and at the end of the course of ECT across 85 regions. Lower panel, left: Effect sizes of right unilateral stimulations were consistently higher on the right side than on the left side. Lower panel, right: Scatter plot of regional EF versus regional volume change (r = 0.38; p <0.001; df = 83; t = 3.77). (d) = Cohen's d effect size..

The online version of this article includes the following source data for figure 1:

**Source data 1.** Mean electric field and volume change in 85 brain regions.

between laterality indices for EF and ΔVol also had a positive relationship (*Figure 2*, t = 2.13, df = 40, r = 0.32, p=0.04) across 42 regions (brain stem is missing, since it is not a bilateral structure).

## Electric field and volume change

In a multiple regression analysis controlled for age, number of ECT sessions and site, we found that left hippocampus and left amygdala had a strong relationship with EF in these regions (FDR corrected p<0.01, *Table 1*). Post hoc analyses of the hippocampus (*Figure 3*) and amygdala (*Figure 4*) illustrate that the relationship between EF and ΔVol was dose-dependent and scaled across the hemispheres (hippocampus: t = 5.97, df = 300, r = 0.3259, p<0.0001; amygdala: t = 11.3538, df = 300, r = 0.5482, p<0.0001). Age was a necessary covariate since it was a confound in our model: both the spatial distribution of EF and volume changes correlate with age (*Deng et al., 2015*). We add number of ECT as a covariate to the model to be able to compare the relative influence of EF and number of ECT on volume change. In both left hippocampus and amygdala the effect size of EF was the largest (hippocampus: $t_{EF}$ = 4.5, $t_{Age}$ = −2.7, $t_{ECTnum}$ = 3.3, amygdala: $t_{EF}$ = 3.9, $t_{Age}$ = −1.1, $t_{ECTnum}$ = 2.1; *Table 1*).

We also investigated the spatial specificity of these correlations. First, we permutated the regional labels in the volumetric changes across all possible ROIs and calculated the correlations between the EF and ΔVol. The correlation between EF and the corresponding ΔVol (*Figure 3—figure supplement 1*, *Figure 4—figure supplement 1*, left panels, indicated with red dot) was always in the top 5% among all possible correlations. Second, we permutated the region labels in the EF across all possible ROIs and calculated the correlations between the EF and ΔVol (*Figure 3—figure*

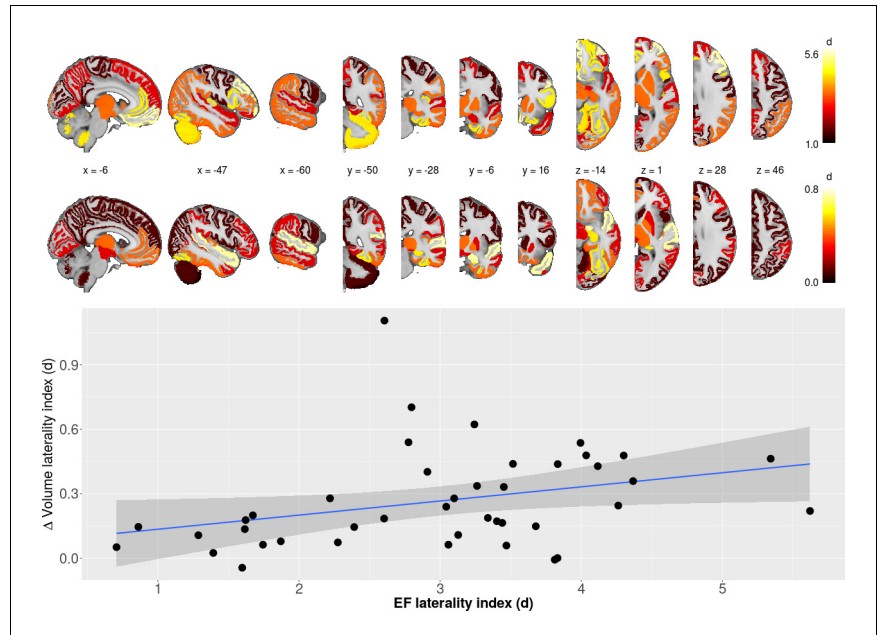

**Figure 2.** Laterality differences in EF and Δvol (upper panel) as well as the relationship between laterality between EF/Δvol (lower panel). Regression line indicates the correlation between laterality indices of EF and volume change (r = 0.32; p<0.05; df = 40; t = 2.13).

The online version of this article includes the following source data for figure 2:

**Source data 1.** Mean electric field and volume change asymteries in corresponding 42 brain regions.

supplement 1, Figure 4—figure supplement 1 right panels). Overall these results indicate a strong spatial selectivity in the relationship between EF and ΔVol.

## Electric field, volume change, and clinical response

We further investigated if EF directly or indirectly (mediated via volume change) leads to clinical response. In a multiple regression analysis, we tested if volumetric changes controlled for age, number of ECT sessions, and site had an effect on clinical response measured by MADRS changes. Results indicated that none of the volume changes across the 85 ROIs had a significant relationship with clinical response (*Supplementary file 3*, hippocampus: $t_{\Delta VOL}$ = 0.2, $t_{Age}$ = 5.4, $t_{ECTnum}$ = −2.7, amygdala: $t_{\Delta VOL}$ = 0.1, $t_{Age}$ = 5.6, $t_{ECTnum}$ = −3.0). These results therefore contradicted the hypothesis that EF by increasing brain volume indirectly exerts its effect on clinical response, given the negative results between the volume change (mediator) and MADRS change (outcome). Testing the direct effect of the EF, we failed to find significant correlations between EF and clinical response (*Supplementary file 4*, hippocampus: $t_{EF}$ = 1.2, $t_{Age}$ = 5.7, $t_{ECTnum}$ = −3.0, amygdala: $t_{EF}$ = 1.1, $t_{Age}$ = 5.7, $t_{ECTnum}$ = −3.0). Similar to earlier studies, age strongly correlated with both clinical response (*Haq et al., 2015*; *O'Connor et al., 2001*), also see *Clinical Results*) and EF distribution (*Deng et al., 2015*), therefore we controlled for age in our model. The rationale for including the number of ECT treatments as covariate needs more explanation. Due to the naturalistic nature of the design, where most sites followed the patient until response or site-determined criteria for ECT discontinuation, we observed a negative relationship between clinical response and the number of ECT treatments. Not controlling for this variable could lead to spurious correlation between volume change and clinical response (for more on this see *Oltedal et al., 2018*). In a post-hoc analysis, we also examined the interaction between EF and volume change in relation to clinical outcome (excluding age as a covariate), but we again failed to find significant effects for any region. To explore further, we investigated if changing age to baseline volume in the mixed model would modify results,

**Table 1.** The relationship between volume changes and EF across individuals ($\Delta$ Vol ~ EF + Age + ECTnum).

| | roi | $t_{EF}$ | $p_{EF}$ | $t_{Age}$ | $t_{ECTnum}$ | $BH^{EF}_{FDR}$ |
|---|---|---|---|---|---|---|
| 1 | $\Delta$ VOL$_{Left.Cerebellum.Cortex}$ | −0.3668 | 0.7143 | −0.1150 | 1.9368 | 0.8205 |
| 2 | $\Delta$ VOL$_{Left.Thalamus.Proper}$ | 0.0244 | 0.9805 | −0.4046 | 2.8696 | 0.9952 |
| 3 | $\Delta$ VOL$_{Left.Caudate}$ | 0.6555 | 0.5132 | −0.8301 | 2.6428 | 0.6924 |
| 4 | $\Delta$ VOL$_{Left.Putamen}$ | 0.5737 | 0.5671 | −0.5992 | 1.3203 | 0.7212 |
| 5 | $\Delta$ VOL$_{Left.Pallidum}$ | 0.0060 | 0.9952 | 0.1026 | 1.2295 | 0.9952 |
| 6 | $\Delta$ VOL$_{Brain.Stem}$ | 1.2114 | 0.2278 | 0.8536 | 1.2309 | 0.4466 |
| 7 | $\Delta$ VOL$_{Left.Hippocampus}$ | 4.5102 | 0.0000 | −2.6814 | 3.3221 | 0.0012 |
| 8 | $\Delta$ VOL$_{Left.Amygdala}$ | 3.9069 | 0.0001 | −1.0572 | 2.1018 | 0.0061 |
| 9 | $\Delta$ VOL$_{Left.Accumbens.area}$ | 2.0238 | 0.0449 | −3.4456 | 1.7246 | 0.1737 |
| 10 | $\Delta$ VOL$_{Left.VentralDC}$ | 0.1740 | 0.8621 | 0.0605 | 2.2614 | 0.9395 |
| 11 | $\Delta$ VOL$_{Right.Cerebellum.Cortex}$ | −0.5564 | 0.5788 | 0.0677 | 1.3212 | 0.7235 |
| 12 | $\Delta$ VOL$_{Right.Thalamus.Proper}$ | 0.4582 | 0.6475 | 0.3541 | 4.0787 | 0.7712 |
| 13 | $\Delta$ VOL$_{Right.Caudate}$ | 1.2293 | 0.2210 | 1.0254 | 1.5097 | 0.4466 |
| 14 | $\Delta$ VOL$_{Right.Putamen}$ | 1.0724 | 0.2854 | −0.5112 | 1.4987 | 0.4756 |
| 15 | $\Delta$ VOL$_{Right.Pallidum}$ | 0.6045 | 0.5465 | 0.8016 | 2.9589 | 0.7181 |
| 16 | $\Delta$ VOL$_{Right.Hippocampus}$ | 1.5090 | 0.1336 | −1.2924 | 3.2473 | 0.3441 |
| 17 | $\Delta$ VOL$_{Right.Amygdala}$ | 2.9945 | 0.0032 | −0.6087 | 4.2603 | 0.0344 |
| 18 | $\Delta$ VOL$_{Right.Accumbens.area}$ | 1.9563 | 0.0524 | −0.8782 | 3.5228 | 0.1937 |
| 19 | $\Delta$ VOL$_{Right.VentralDC}$ | 0.3488 | 0.7278 | 0.5197 | 0.7438 | 0.8248 |
| 20 | $\Delta$ VOL$_{ctx.lh.bankssts}$ | 1.1757 | 0.2417 | −0.4102 | 2.5801 | 0.4466 |
| 21 | $\Delta$ VOL$_{ctx.lh.caudalanteriorcingulate}$ | 1.3404 | 0.1823 | −1.2881 | 2.2330 | 0.4254 |
| 22 | $\Delta$ VOL$_{ctx.lh.caudalmiddlefrontal}$ | −1.8989 | 0.0596 | −0.3804 | 2.0087 | 0.2112 |
| 23 | $\Delta$ VOL$_{ctx.lh.cuneus}$ | 0.9827 | 0.3274 | 0.1037 | 2.0348 | 0.5352 |
| 24 | $\Delta$ VOL$_{ctx.lh.entorhinal}$ | 3.2229 | 0.0016 | −1.2447 | 1.6659 | 0.0335 |
| 25 | $\Delta$ VOL$_{ctx.lh.fusiform}$ | 3.0717 | 0.0026 | −0.1806 | 2.1319 | 0.0344 |
| 26 | $\Delta$ VOL$_{ctx.lh.inferiorparietal}$ | 1.5131 | 0.1325 | 0.8515 | 2.3077 | 0.3441 |
| 27 | $\Delta$ VOL$_{ctx.lh.inferiortemporal}$ | 2.6985 | 0.0078 | 0.6415 | 1.9131 | 0.0577 |
| 28 | $\Delta$ VOL$_{ctx.lh.isthmuscingulate}$ | −0.3275 | 0.7438 | −0.4344 | 2.9060 | 0.8319 |
| 29 | $\Delta$ VOL$_{ctx.lh.lateraloccipital}$ | 1.1916 | 0.2354 | 0.3669 | 1.2752 | 0.4466 |
| 30 | $\Delta$ VOL$_{ctx.lh.lateralorbitofrontal}$ | 1.4274 | 0.1557 | −0.0081 | 1.5758 | 0.3780 |
| 31 | $\Delta$ VOL$_{ctx.lh.lingual}$ | 0.1391 | 0.8896 | 0.3506 | 2.4745 | 0.9572 |
| 32 | $\Delta$ VOL$_{ctx.lh.medialorbitofrontal}$ | 1.0744 | 0.2845 | −0.1246 | 1.1852 | 0.4756 |
| 33 | $\Delta$ VOL$_{ctx.lh.middletemporal}$ | 2.0679 | 0.0405 | −0.3780 | 2.2600 | 0.1720 |
| 34 | $\Delta$ VOL$_{ctx.lh.parahippocampal}$ | 1.2683 | 0.2068 | −0.2446 | 2.8373 | 0.4466 |
| 35 | $\Delta$ VOL$_{ctx.lh.paracentral}$ | −2.0829 | 0.0391 | 0.2511 | 4.0937 | 0.1720 |
| 36 | $\Delta$ VOL$_{ctx.lh.parsopercularis}$ | −0.6949 | 0.4883 | −0.7822 | 1.8435 | 0.6694 |
| 37 | $\Delta$ VOL$_{ctx.lh.parsorbitalis}$ | 0.8057 | 0.4218 | −1.0427 | 0.9524 | 0.6289 |
| 38 | $\Delta$ VOL$_{ctx.lh.parstriangularis}$ | 0.8228 | 0.4120 | −1.2157 | 2.7977 | 0.6254 |
| 39 | $\Delta$ VOL$_{ctx.lh.pericalcarine}$ | 0.4426 | 0.6587 | −0.0479 | 1.8463 | 0.7712 |
| 40 | $\Delta$ VOL$_{ctx.lh.postcentral}$ | 0.8692 | 0.3862 | −1.7655 | 2.5145 | 0.5969 |
| 41 | $\Delta$ VOL$_{ctx.lh.posteriorcingulate}$ | −0.8698 | 0.3859 | −0.6961 | 3.3193 | 0.5969 |
| 42 | $\Delta$ VOL$_{ctx.lh.precentral}$ | −0.7279 | 0.4679 | −1.2884 | 2.4234 | 0.6682 |
| 43 | $\Delta$ VOL$_{ctx.lh.precuneus}$ | −1.5879 | 0.1145 | −0.4353 | 3.6729 | 0.3441 |
| 44 | $\Delta$ VOL$_{ctx.lh.rostralanteriorcingulate}$ | 1.3315 | 0.1852 | −0.4449 | 0.5630 | 0.4254 |

*Table 1 continued on next page*

*Table 1 continued*

| | roi | $t_{EF}$ | $p_{EF}$ | $t_{Age}$ | $t_{ECTnum}$ | $BH^{EF}_{FDR}$ |
|---|---|---|---|---|---|---|
| 45 | Δ VOL$_{ctx.lh.rostralmiddlefrontal}$ | −0.7192 | 0.4732 | −1.6205 | 1.1936 | 0.6682 |
| 46 | Δ VOL$_{ctx.lh.superiorfrontal}$ | −1.2073 | 0.2293 | −0.5851 | 2.1065 | 0.4466 |
| 47 | Δ VOL$_{ctx.lh.superiorparietal}$ | −1.7423 | 0.0836 | 0.6952 | 3.3288 | 0.2734 |
| 48 | Δ VOL$_{ctx.lh.superiortemporal}$ | 2.2820 | 0.0240 | −2.0868 | 1.6393 | 0.1199 |
| 49 | Δ VOL$_{ctx.lh.supramarginal}$ | 0.5717 | 0.5685 | −0.2467 | 2.1282 | 0.7212 |
| 50 | Δ VOL$_{ctx.lh.frontalpole}$ | −0.2029 | 0.8395 | −0.2904 | 0.4776 | 0.9267 |
| 51 | Δ VOL$_{ctx.lh.temporalpole}$ | 2.5288 | 0.0125 | −0.0731 | 1.3167 | 0.0762 |
| 52 | Δ VOL$_{ctx.lh.transversetemporal}$ | 0.4387 | 0.6616 | −0.4617 | 2.1817 | 0.7712 |
| 53 | Δ VOL$_{ctx.rh.bankssts}$ | 0.1121 | 0.9109 | 2.0777 | 2.9991 | 0.9678 |
| 54 | Δ VOL$_{ctx.rh.caudalanteriorcingulate}$ | −1.4295 | 0.1551 | 1.2935 | 2.4016 | 0.3780 |
| 55 | Δ VOL$_{ctx.rh.caudalmiddlefrontal}$ | −2.9569 | 0.0036 | 1.6943 | 2.6065 | 0.0344 |
| 56 | Δ VOL$_{ctx.rh.cuneus}$ | −0.0087 | 0.9930 | −1.1806 | 2.4017 | 0.9952 |
| 57 | Δ VOL$_{ctx.rh.entorhinal}$ | 1.2514 | 0.2129 | 0.7897 | 2.4722 | 0.4466 |
| 58 | Δ VOL$_{ctx.rh.fusiform}$ | 1.5380 | 0.1263 | 0.7997 | 4.7854 | 0.3441 |
| 59 | Δ VOL$_{ctx.rh.inferiorparietal}$ | −2.9902 | 0.0033 | 1.6520 | 0.7114 | 0.0344 |
| 60 | Δ VOL$_{ctx.rh.inferiortemporal}$ | 0.9300 | 0.3540 | 1.9310 | 3.3455 | 0.5677 |
| 61 | Δ VOL$_{ctx.rh.isthmuscingulate}$ | 0.0325 | 0.9741 | 0.4230 | 1.1493 | 0.9952 |
| 62 | Δ VOL$_{ctx.rh.lateraloccipital}$ | 1.1796 | 0.2401 | 0.6095 | 1.5161 | 0.4466 |
| 63 | Δ VOL$_{ctx.rh.lateralorbitofrontal}$ | 0.5347 | 0.5937 | 0.3393 | 2.9240 | 0.7314 |
| 64 | Δ VOL$_{ctx.rh.lingual}$ | −0.0753 | 0.9401 | −1.9555 | 3.5258 | 0.9865 |
| 65 | Δ VOL$_{ctx.rh.medialorbitofrontal}$ | 0.7090 | 0.4795 | 1.5479 | 2.3419 | 0.6682 |
| 66 | Δ VOL$_{ctx.rh.middletemporal}$ | −0.6005 | 0.5492 | 2.1275 | 3.6781 | 0.7181 |
| 67 | Δ VOL$_{ctx.rh.parahippocampal}$ | 1.5217 | 0.1303 | 0.5057 | 3.1874 | 0.3441 |
| 68 | Δ VOL$_{ctx.rh.paracentral}$ | −3.5101 | 0.0006 | 2.1809 | 2.2718 | 0.0170 |
| 69 | Δ VOL$_{ctx.rh.parsopercularis}$ | −2.5585 | 0.0116 | 2.8854 | 2.9459 | 0.0756 |
| 70 | Δ VOL$_{ctx.rh.parsorbitalis}$ | 1.0872 | 0.2788 | −0.5812 | 2.3737 | 0.4756 |
| 71 | Δ VOL$_{ctx.rh.parstriangularis}$ | −1.2468 | 0.2146 | 1.0686 | 2.6086 | 0.4466 |
| 72 | Δ VOL$_{ctx.rh.pericalcarine}$ | 1.5878 | 0.1146 | −0.0096 | 2.2815 | 0.3441 |
| 73 | Δ VOL$_{ctx.rh.postcentral}$ | −1.7565 | 0.0812 | 1.2943 | 3.0605 | 0.2734 |
| 74 | Δ VOL$_{ctx.rh.posteriorcingulate}$ | −1.5171 | 0.1315 | 2.0716 | 1.4731 | 0.3441 |
| 75 | Δ VOL$_{ctx.rh.precentral}$ | −2.4918 | 0.0139 | 0.9967 | 3.7013 | 0.0762 |
| 76 | Δ VOL$_{ctx.rh.precuneus}$ | −2.0231 | 0.0450 | −0.1921 | 2.5419 | 0.1737 |
| 77 | Δ VOL$_{ctx.rh.rostralanteriorcingulate}$ | 2.2083 | 0.0288 | 1.3734 | 2.3606 | 0.1362 |
| 78 | Δ VOL$_{ctx.rh.rostralmiddlefrontal}$ | −2.6842 | 0.0081 | 0.5804 | 2.2235 | 0.0577 |
| 79 | Δ VOL$_{ctx.rh.superiorfrontal}$ | −3.0013 | 0.0032 | 1.1011 | 3.2699 | 0.0344 |
| 80 | Δ VOL$_{ctx.rh.superiorparietal}$ | −2.7495 | 0.0067 | 0.9014 | 2.0779 | 0.0574 |
| 81 | Δ VOL$_{ctx.rh.superiortemporal}$ | 0.4377 | 0.6623 | 1.2455 | 4.4002 | 0.7712 |
| 82 | Δ VOL$_{ctx.rh.supramarginal}$ | −2.4794 | 0.0143 | 2.7408 | 3.0429 | 0.0762 |
| 83 | Δ VOL$_{ctx.rh.frontalpole}$ | 1.1256 | 0.2623 | −0.1784 | 1.9185 | 0.4644 |
| 84 | Δ VOL$_{ctx.rh.temporalpole}$ | 0.7274 | 0.4682 | 0.5099 | 3.7696 | 0.6682 |
| 85 | Δ VOL$_{ctx.rh.transversetemporal}$ | 1.1426 | 0.2551 | 0.6448 | 3.2405 | 0.4614 |

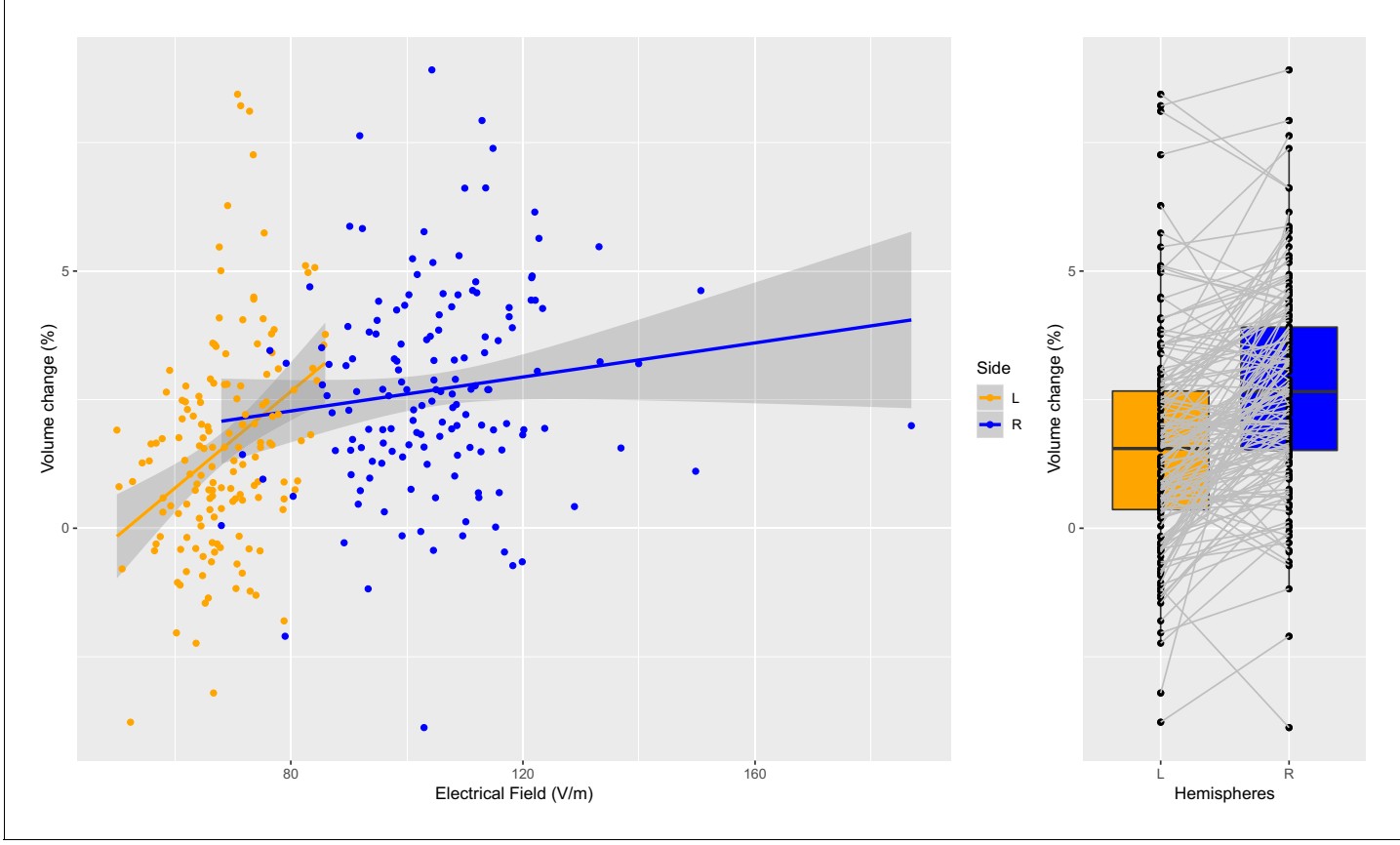

**Figure 3.** Individual specific relationship between EF and volume change in the hippocampus. Left: Scatterplot of EF versus volume change in the hippocampus (t = 5.97, df = 300, r = 0.33, p < 0.0001, left and right side together). There is a significant relationship on the left side (orange dots; t = 4.53, df = 149, r = 0.35, p < 0.0001), but not on the right side (probably due to ceiling effect) (t = 1.59, df = 149, r = 0.13, p = 0.11). Right: The difference in right and left hippocampal volume changes is significant (t = 7.76, df = 150, mean difference = 0.011, p < 0.0001).

The online version of this article includes the following source data and figure supplement(s) for figure 3:

**Source data 1.** Left and right hippocampal EF and volume change in 151 individual.
**Figure supplement 1.** Hippocampal EF and volume change.
**Figure supplement 1—source data 1.** Hippocampal EF and volume change: permutation values.

but we did not find significant effects (age and baseline volume correlates strongly across almost all regions – **Supplementary file 5**).

## Discussion

This study investigated the relationship between electric field, volume change and clinical response to ECT. We used a large sample of subjects with depression receiving ECT with right unilateral electrode placement from the GEMRIC database. The key findings included a lateralization (right >left) of the electric field and changes in regional brain volume in association with ECT. The use of right unilateral electrode placement, which elicits greater right hemisphere electric fields, can thus be dissociated from generalized seizure activity such that their contributions to antidepressant mechanisms may be at least partially distinct. Further, regional volume increase and electric field distributions were strongly related, especially in the left hippocampus and left amygdala. Here, the observed relationships between electric field and volume change suggest that a minimum electric field of 30–40 V/m is necessary for subsequent changes in brain structure, and that EF may have a 'ceiling effect' above approximately 100 V/m as illustrated for right hippocampal volume (see **Figure 3**). However, volume change and electric field were not statistically related to clinical response after controlling for age, number of ECT sessions and site. Below, we discuss potential mechanisms for electric field

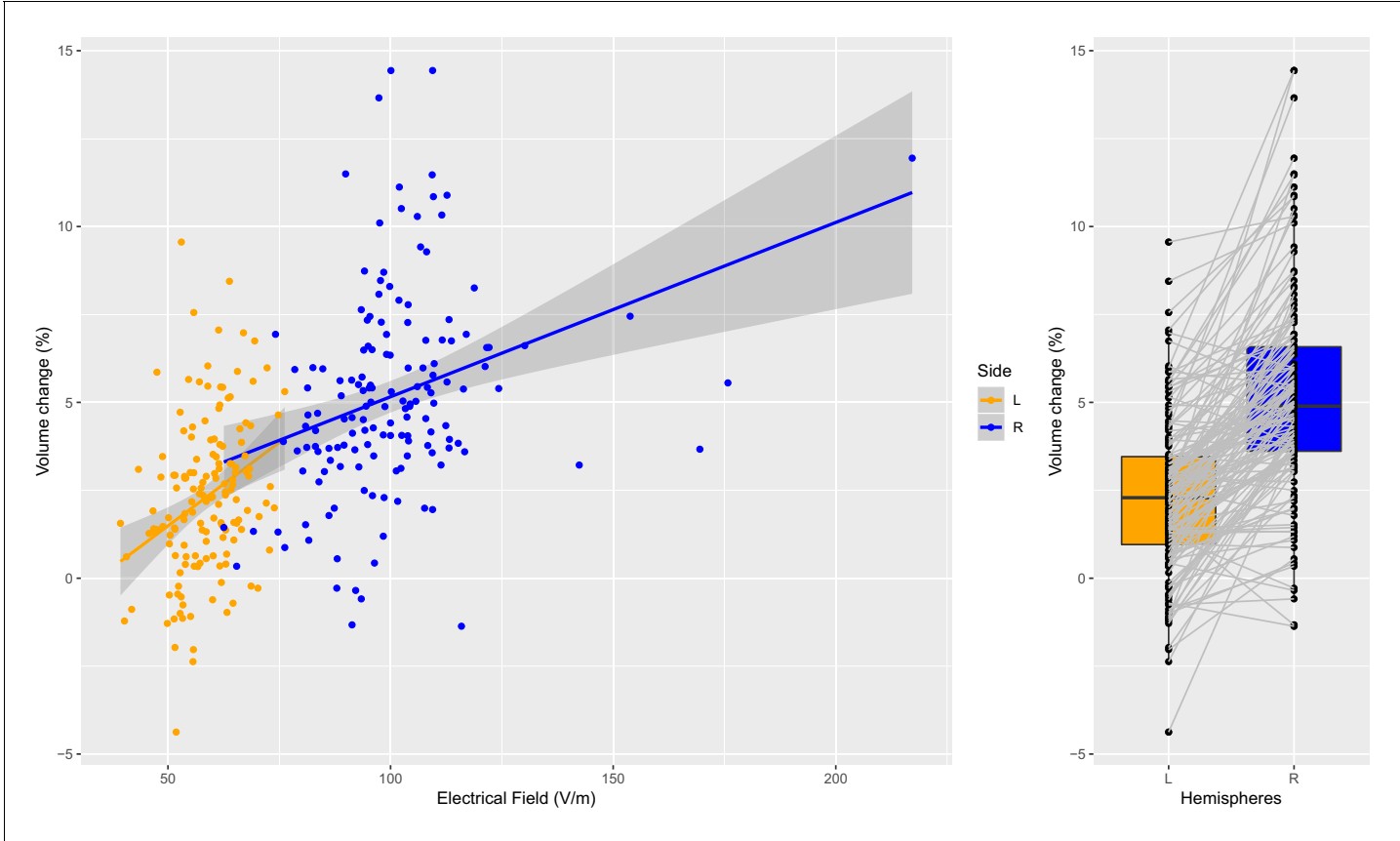

**Figure 4.** Individual specific relationship between EF and volume change in the amygdala. Left: Scatterplot of EF versus volume change in the amygdala (t = 11.35, df = 300, r = 0.55, p<0.0001; left and right side together). Both the left (orange dots) and right (blue dots) hemisphere shows highly significant relationships (t = 4.01, df = 149, r = 0.31, p=0.0001; and t = 4.02, df = 149, r = 0.31, p=0.0001). Right: The difference in right and left amygdala volume changes is significant (t = 13.58, df = 150, mean difference = 0.029, p<0.0001).

The online version of this article includes the following source data and figure supplement(s) for figure 4:

**Source data 1.** Left and right amygdala EF and volume change in 151 individual.
**Figure supplement 1.** Amygdala EF and volume change.
**Figure supplement 1—source data 1.** Amygdala EF and volume change: permutation values.

and volume change that may be considered both independent and synergistic with seizure activity. We also discuss potential future directions to elucidate the role of electric field distributions with clinical response.

The biological underpinnings of ECT-mediated volume change (*Oltedal et al., 2018*; *Ousdal et al., 2019*) could be related to seizure activity, cerebral blood flow, electric field strength, or synergy between the generalized seizure and electric field (e.g. the electric field determines site and focality of seizure initiation, which can subsequently affect seizure propagation and termination). Several neuroplastic mechanisms including neurogenesis, angiogenesis, synaptogenesis, gliogenesis may be specific to the rapidly changing electric field (*Bouckaert et al., 2014*; *Tang et al., 2017*). Although heavily debated (*Sorrells et al., 2018*; *Boldrini et al., 2018*; *Andreae, 2018*), the support for adult neurogenesis is based on pre-clinical studies demonstrating neuronal division and differentiation related to suprathreshold electric stimulation (*Scott et al., 2000*; *Madsen et al., 2000*; *Perera et al., 2007*; *Segi-Nishida, 2011*). However, neurogenesis as the sole mechanism of neuroplasticity may be incompatible with the time frame and expansive volume change. Specifically, the ECT series is less than one month in duration, but pre-translational investigations have established that adult neurogenesis may take up to six months (*Kohler et al., 2011*). Furthermore, adult neurogenesis is limited to the hippocampus and olfactory bulb and does not support the volume change in 82 out of 85 regions demonstrated in our investigation (*Kornack and Rakic, 1999*). Alternatively,

volume change may be related to fluid shifts due to vascularization (*Hellsten et al., 2004*), blood flow changes (*Milo et al., 2001*; *Leaver et al., 2019*) and inflammation (*Wennström et al., 2004*; *Jansson et al., 2009*; *Fluitman et al., 2011*; *van Buel et al., 2015*; *Yrondi et al., 2018*). Vasogenic edema secondary to the hypertensive surge commonly associated with electroconvulsive stimulation and possible breach of the blood brain barrier could be a potentially iatrogenic mechanism of volumetric increase, but the available pre-clinical and ECT-imaging studies (focused on T2 relaxtion time) so far have produced mixed results (*Andrade and Bolwig, 2014*; *Kunigiri et al., 2007*; *Bolwig et al., 1977*; *Nordanskog et al., 2010*; *Takamiya et al., 2018*). The generalized seizure and global changes in blood flow would not explain the laterality of volumetric changes (right >left) ipsilateral to the hemisphere of stimulation as seen in our current and previous investigations (*Abbott et al., 2014*; *Dukart et al., 2014*; *Pirnia et al., 2016*; *Bouckaert et al., 2016*; *Sartorius et al., 2016*; *Cano et al., 2018*). The laterality with electric field and volume change suggest a mechanistic role of the electric field that may be independent or synergistic with seizure generation. Pre-translational investigations have demonstrated that increased stimulus charge increased dendritic arborization in a dose-related fashion (*Smitha et al., 2014*). Furthermore, the behavioral improvement after electroconvulsive stimulation are related to increased dendritic complexity, synaptic remodeling, and neuronal survival (*Jonckheere et al., 2018*). However, additional pre-clinical studies are clearly needed to resolve the mechanistic link between electric field and neuroplasticity.

Our original hypothesis was that a) local electric field had a causal role in clinical outcome and that b) the corresponding volume change was mediating this relationship. In order to support this model, data analysis should have indicated 1) a significant correlation between volume change and electric field, 2) a significant correlation between clinical outcome and volume change, and 3) that only the effect of volume change is significant in a multilinear regression model when both electric field and volume change is added as covariates (outcome ~volume change + electric field). However, since volume change showed no correlation with clinical change, neither in this dataset, nor in the recently published broader dataset with more heterogeneous ECT electrode placement (*Ousdal et al., 2019*), only the first half of this model, namely that electric field strength was associated with volume change, was supported by our data.

The null relationship between electric field, volume change and clinical outcome may be attributed to demographic (age) and other treatment related factors (number of sessions, rate of

**Table 2.** Clinical and demographics summary.

*Table 2A* Overall Summary

| Site | N | Age (sd) | Medications (med. free, SSRI/SNRI, TCA, AP*) | Average number of ECT | Baseline MADRS | Δ MADRS (%) |
|------|-----|-----------|--------------------------------|-----------|-----------|-----------|
| All | 151 | 57.5 (17.1) | 69,65,10,62 | 10.6 | 33.9 | 61.3 |
| Female | 92 | 56.4 (18.4) | 42,36,8,42 | 10.4 | 34.4 | 63.4 |
| Male | 59 | 59.3 (14.7) | 27,29,2,20 | 10.9 | 33.3 | 58.1 |

*Table 2B* Site Summary

| Site | N | Age (mean) | Age (sd) | | | Baseline MADRS | Δ MADRS (%) |
|------|-----|-----------|---------|--|--|-----------|-----------|
| 1 | 30 | 39.87 | 12.68 | | | 40.73 | 45.12 |
| 2 | 33 | 64.48 | 8.93 | | | 31.36 | 69.48 |
| 3 | 16 | 73.62 | 12.45 | | | 29.56 | 77.24 |
| 4 | 23 | 46.87 | 9.19 | | | 29.96 | 43.18 |
| 5 | 2 | 62.50 | 0.71 | | | 36.75 | 32.03 |
| 6 | 18 | 48.50 | 16.77 | | | 33.83 | 57.12 |
| 7 | 29 | 72.66 | 7.57 | | | 35.07 | 79.13 |

*med. free: medication free, SSRI: selective serotonin reuptake inhibitor, SNRI: serotonin and norepinephrine reuptake inhibitors, TCA: tricyclic antidepressants, AP: antipsychotic medications, there were not patients on MAO inhibitors.

response). For example, age-related structural brain changes may mediate these relationships and thus were an important consideration in our analysis. Our results are consistent with previous ECT investigations demonstrating that older patients often have higher response rates (*O'Connor et al., 2001*; *Nordenskjöld et al., 2012*; *Brus et al., 2017*). Previous electric field modeling investigations have demonstrated that age-related structural brain changes modulate the spatial distribution of the calculated electric field (*Deng et al., 2009*). However, when including age in the assessment with electric field, volume change and clinical outcome, our results suggest more complex or alternative mechanisms underlie differential age-related response to ECT.

Additionally, it was necessary to control in our regression models for the number of ECT treatments. In our earlier paper (*Oltedal et al., 2018*) we found a mild effect between hippocampus volume change and clinical response, but, counterintuitively, increased volume change was associated with worse outcomes. However, this relationship was completely absent when we controlled for the number of ECTs. We have previously demonstrated a dose-response relationship between hippocampal volume change and the number of ECT sessions (*Oltedal et al., 2018*). Also, due to the naturalistic design, clinical outcome correlated with the number of ECT sessions: patients with the worse or slower response received more ECT treatments. Mediation analysis supported a very similar situation in our sample with p=0.035 and p=0.034 in L and R Hippocampus reflectively (Sobel test).

It was, therefore, necessary to control for the number of ECT sessions to avoid detecting spurious correlations between clinical response and volume change. Without an earlier, fixed mid-point assessment, we are unable to assess differences in rate of change, which could be relevant to specific depression subtypes (*Drysdale et al., 2017*) and eventual clinical response. Notably, the overall volume changes measured in this study do not permit us to make conclusions about more structure-function relationships that might be better assessed with shape or hippocampal subfield analysis (*Roddy et al., 2019*; *Takamiya et al., 2019*).

Finally, the volume change required for response may be non-linear. A minimum electric field of 30–40 V/m may be necessary to induce neuroplasticity. Increasing the electric field between 30–40 V/m and 100 V/m is related to a monotonic increase in hippocampal volume. Electric field above 100 V/m is still associated with hippocampal neuroplasticity but the dose-response relationship may be less robust and represent a ceiling effect of electric-field induced neuroplasticity as illustrated in *Figure 3*. Surpassing the neuropolasticity threshold (100 V/m) appears to be unrelated to further volumetric increases and antidepressant response. Thus, the relationship between e-field and volumetric changes may be conceptualized as a 'neuroplasticity threshold' between 30–40 V/m and 100 V/m. This thresholding effect also preserves the laterality of electric field and neuroplasticity. Our sample was limited to right unilateral electrode placement. In the left hippocampus, the maximum electric field is ~80 V/m (*Figure 3*) and below the 100 V/m 'ceiling effect' noted in the right hemisphere. Consequently, in our right unilateral sample, hippocampal electric field and related changes in neuroplasticity will demonstrate laterality effects.

Our findings indicate widespread and robust volume changes in both cortical and subcortical regions. The GEMRIC group recently published a comprehensive paper on a larger dataset with similar volumetric findings. The processing pipeline that was used has been validated against many commonly used tools for estimating longitudinal volume change (*Holland et al., 2012*; *Holland et al., 2011*). Specifically, it was previously compared head-to-head with FreeSurfer 5.3, and we have already repeated this comparison for data from one of the GEMRIC sites (*Oltedal et al., 2017*). Our comparisons of power estimations based on results from the FreeSurfer longitudinal pipeline and Quarc (Table 3 in *Oltedal et al., 2017*) were in line with those of the earlier publications. In agreement with previous research, the effect sizes show regional differences indicating that previous studies with smaller sample sizes were underpowered to detect cortical changes, and that can explain why they only found subcortical volume increase. Furthermore, using the same methodology, we did not find any significant volume change in the 95 healthy controls (received no ECT) who were imaged at two time points (*Ousdal et al., 2019*).

We acknowledge several limitations that influence result interpretation. First, our approach was agnostic to seizure duration, which may contribute to the effects of EF on regional volumes and clinical response. This investigation also does not preclude the possible role of seizure in both volume changes and clinical outcomes. However, the selection of right unilateral electrode placement subjects does attempt to disentangle the impact of the generalized seizure from the lateralized electric

field. Second, the electric field models a single current pulse and ignores the temporal dynamics of stimulus (pulse-width, and frequency and duration of the pulse train) (*Swartz, 2006*; *Swartz et al., 2012*). Differences in pulse width, for example, may affect volume change and clinical outcomes. Furthermore, differences in maximal charge, unrelated to current amplitude, are different between the US and Europe (Europe permits twice the US maximal charge). The analysis included patients treated with one of two different ECT devices. We controlled for the differences in current related to the two devices with the electric field modeling (800mA for the spECTrum, 900mA for the Thymatron), but we are unable to control for other differences in stimulus delivery related to the device. Third, the study sites in this mega-analysis likely include heterogeneity in patient selection and other treatment related factors that were not controlled. Despite these site differences, the large sample size and additional inclusion criteria permitted whole brain analyses with electric field, and within-subject volume change and clinical outcomes. Finally, we did not assess cognitive correlates with electric field or volume change. General clinical experience and previous results from studies investigating electrode placement strategies indicate that ECT-mediated neurocognitive side effects are influenced by electrode placement (*d'Elia, 1970*; *Sackeim et al., 2000*; *Kolshus et al., 2017*). Previous electric field studies on simulated head models have already shown that cognitive side effects might be attributed to the electric field spatial distributions associated with different electrode placements (*Bai et al., 2017*; *Deng et al., 2011*). These considerations would indicate that these volumetric changes might be associated with cognitive side-effects, but further studies are needed to confirm this relationship.

## Conclusion

This investigation is the first demonstration that the ECT-induced electric field is related to increases in cortical and subcortical structures. These results support that the electric field, independent or synergistic with seizure activity and other stimulation parameters, can have a profound effect on the biology of the human brain. The electric path originates from the ECT electrode handle, which delivers a constant stimulus current from the scalp. From the scalp, the electric path travels through skin, skull, cerebral spinal fluid, and brain. Each tissue type has different conductive properties and abundant individual variability (*Deng et al., 2015*). This variability creates different electric field doses despite the similar current at the scalp. These differences in current may lead to both differences in volume changes as well as clinical outcomes. In our investigation, the electric field-induced volume change in the bilateral amygdala and the left hippocampus suggests regional specificity, but the association of these volumetric changes with clinical outcomes remains elusive. Better controlled prospective trials are needed to answer if these robust volume changes and corresponding electric field distributions are associated with any clinical or cognitive consequences.

# Materials and methods

## Participants

GEMRIC is a multi-site consortium focused on improving and individualizing ECT by researching the still elusive mechanisms of action and response-related biomarkers (*Oltedal et al., 2017*). Patients in the GEMRIC database participated in clinical and imaging assessments pre- and post-ECT series. To control for the differential effects of electrode placement on electric fields, we only included subjects who received high-dose (six times the seizure threshold) right unilateral electrode placement throughout the ECT series. We screened 281 patients from 10 sites (*Oltedal et al., 2018*), and data were included from 7 GEMRIC sites with the RUL only criteria (n = 151, 92 F, age: 57.5 ± 17.1, 12 with bipolar depression, 139 with major depression, demographic summary is in *Table 2A and B*). Depression severity was assessed with the Montgomery–Åsberg Depression Rating Scale (MADRS) (*Montgomery and Asberg, 1979*) or 17- or 24-item Hamilton Depression Rating Scale (HAM-D) (*Hamilton, 1960*). For sites collecting only the 17- or 24-item HAM-D, a validated equation was used to convert the 17-item HAM-D to a MADRS score (*Heo et al., 2007*). Clinical response was estimated as the percentage change of the MADRS scores ($\Delta$MADRS = (MADRS$_{TP1}$-MADRS$_{TP2}$)/ MADRS$_{TP1}$). Although more conservative than absolute change or post-ECT depression outcomes (*Vickers, 2001*), the rationale for the use of the proportional change score was to control for the variability of the pre-ECT MADRS. The range of the number of sessions for the ECT series was between

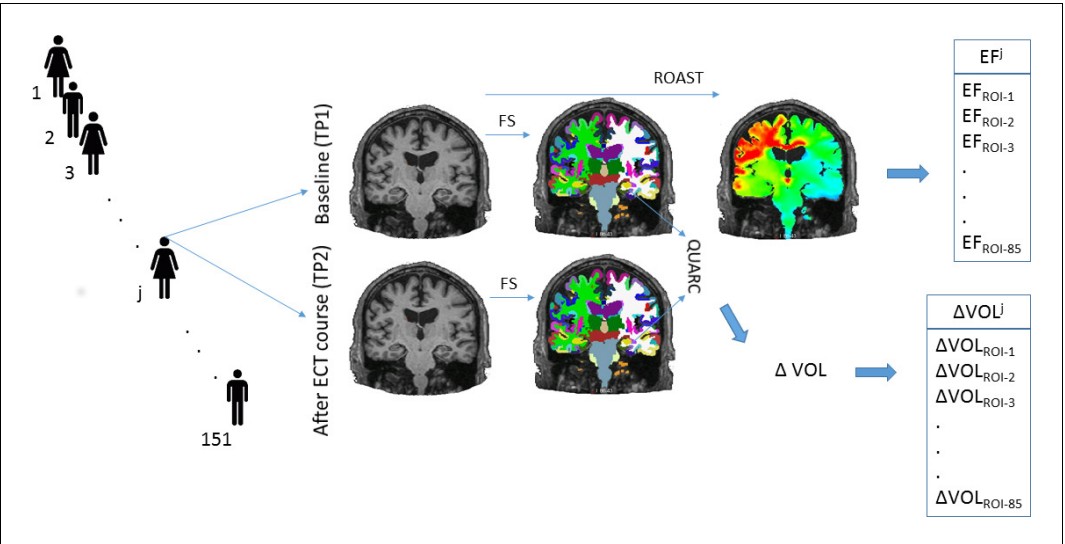

**Figure 5.** Illustration of the methods. We analyzed longitudinal structural MRI data from 151 individuals. We calculated the volume change and the magnitude of electrical field in 85 regions across the human cortex and subcortical structures.

7 and 20. Half of the subjects were medication free during the ECT series (n = 69). Concurrent pharmacotherapy for the remaining subjects was as follows: selective serotonin reuptake inhibitors (SSRI, n = 28), serotonin norepinephrine reuptake inhibitors (SNRI, n = 37), tricyclic antidepressants (TCA, n = 10), and no record of concurrent medications (n = 6). Only five subjects received medication changes during the ECT series (two medication free subjects started SSRI and TCA, one subject switched from SNRI to TCA, one from SNRI to TCA and one from SSRI to SNRI). The results did not change if we used medication status or diagnosis (bipolar or unipolar depression) as a nuisance variable in the linear models of this study. All sites' contributing data (*Table 2B*) received approval by their local ethical committees or institutional review board, and the centralized mega-analysis was approved by the Regional Ethics Committee South-East in Norway (2013/1032 ECT and Neuroradiology, June 1, 2015).

## Imaging

The image processing methods have been detailed previously (*Oltedal et al., 2018*; *Oltedal et al., 2017*). In brief, the sites provided longitudinal 3T T1-weighted MRI images (at baseline and after the end of the course of ECT) with a minimal resolution of 1.3 mm in any direction (detailed parameters in *Supplementary file 6*). The raw DICOM images were uploaded and analyzed on a common server at the University of Bergen, Norway. To guarantee reproducibility, in addition to the common platform, the processing pipelines were implemented in a docker environment (*Merkel, 2014*). First, images were corrected for scanner-specific gradient-nonlinearity (*Jovicich et al., 2006*). Further processing was performed with FreeSurfer version 5.3, which includes segmentation of subcortical structures (*Fischl et al., 2002*) and automated parcellation of the cortex (*Desikan et al., 2006*). In addition to brainstem and bilateral cerebellum, this automated process identified 33 cortical and eight subcortical regions in each hemisphere. Altogether this resulted in 85 regions of interest (ROIs) (*Supplementary file 1*). Next Quarc (*Holland et al., 2011*) was used for unbiased, within-subject assessment of estimation of longitudinal volume change (ΔVol - %) (*Figure 5*). In summary, we calculated bias-free estimation of volumetric change from 85 brain regions across the timespan of an ECT course in 151 individuals who received between 4 to 20 ECT sessions (1 ½ week to 2 month).

## Electric Field modeling

We estimated ECT-induced electric fields with Realistic Volumetric-Approach to Stimulate Transcranial Electric Stimulation (ROAST v1.1) (*Huang et al., 2017*). After segmentation of the structural MRI T1-weighted images, ROAST builds a three-dimensional tetrahedral mesh model of the head. The

segmentation identifies five tissue types: white and gray matter of the brain, cerebrospinal fluid, skull, and scalp, and assigns them different conductivity values: 0.126 S/m, 0.276 S/m, 1.65 S/m, 0.01 S/m, and 0.465 S/m respectively. ECT electrodes of 5 cm diameter were placed over the C2 and FT8 EEG (10–20 system) sites. Study sites from the GEMRIC database used either the Thymatron (Somatics, Venice, Florida, six sites, N = 121) or spECTrum (MECTA Corp., Tualatin, Oregon, one site, N = 30) devices. The electric field was solved using the finite-element method with unit current on the electrodes and, subsequently, it was scaled to the current amplitude of the specific devices (Thymatron 900 mA, spECTrum 800 mA). These procedures resulted in a voxel-wise electric field distribution map in each individual (*Figure 5*) based on the Freesurfer parcellations and segmentations. The voxel values with the top and lowest one percentile in each ROI were omitted during calculations to reduce boundary effects.

## Statistical analysis

### Laterality of electric field and volume change

Our statistical analysis was performed in R (*R Development Core Team, 2013*), and the underlying analyses can be found at https://github.com/argyelan/Publications/ (copy archived at https://github.com/elifesciences-publications/Publications-1) in org mode (*Schulte et al., 2012*). We first calculated the effect sizes (Cohen's d) for longitudinal volume changes in each region. We assessed the correlations between the average electric fields and the effect sizes of volume changes across all the 85 regions. We further explored the hemispheric differences by calculating the pair-wise difference in volume changes across the corresponding ROIs (42 pairs). We defined the laterality index as the effect size of the pair-wise difference for both EF and ΔVol among homotopic ROIs. We then assessed the correlations between laterality indices of EF and ΔVol across the 42 pairs of regions.

## Electric field and volume change

We assessed the relationship between EF and Δ Vol with the following linear mixed effect model in all 85 regions: 1) ΔVol ~EF + Age + number of ECT sessions + site (where EF, age, and number of ECT sessions were fixed effects, and site was random effect, while the dependent variable was volume change). Age, number of ECT sessions, and site, considered as nuisance variables, were included based on our prior observations of an inverse relationship between ECT session number and response (*Oltedal et al., 2018*). Further, age is also shown to impact clinical response (older patients have increased probability of response, in our sample: t = 5.75, df = 149, r = 0.43, $p<10^{-7}$) and age-related changes on brain structure are related to EF (*Deng et al., 2015*). We used Benjamini and Hochberg false discovery rate (FDR) correction method (*Benjamini and Hochberg, 1995*) to control for multiple comparisons across 85 ROIs, where a conservative FDR -corrected p<0.01 was chosen as the statistical threshold of significance.

## Electric field, volume change, and clinical response

We assessed the relationship between ΔMADRS and EF and Δ Vol with the following linear mixed effect model in all 85 regions: 1) ΔMADRS ~ ΔVol + Age + number of ECT sessions + site; and 2) ΔMADRS ~EF + Age + number of ECT sessions + site (site as random effect). We used the same Benjamini and Hochberg FDR correction for multiple comparison corrections. In addition to analyzing the percentage change of the clinical response, we also evaluated the same models with absolute changes, using baseline MADRS as a covariate. We provided the results of these analyses in the second half of the corresponding Supplementary Files.

## Acknowledgements

This work was supported by the Western Norway Regional Health Authority (#911986 to KJO and # 912238 to LO) and the University of Bergen, Norway, the Fulbright Program (to LO); the National Institute of Mental Health (#MH092301, #MH110008, #MH102743) (to KN and RE), #MH119616 (to MA), #MH111826 (to CA) and the German Research Foundation (DFG FOR2107, DA1151/5-1 and DA1151/5-2; SFB-TRR58, Projects C09 and Z02) (UD) and Innovative medical research (RE111604,

RE111722) (to RR). ZD is supported by the National Institute of Mental Health Intramural Research Program (ZIAMH00295).

We would like to acknowledge the logistic and academic support of the whole GEMRIC consortium outside of the authorship of this manuscript. These collaborators were not only actively involved in this manuscript but their valuable feedback on our regular meetings contributed to this manuscript in many levels. The full overview of the GEMRIC board members can be find here: https://helse-bergen.no/en/avdelinger/psykisk-helsevern/forskingsavdelinga-divisjon-psykisk-helsevern/gemric-the-global-ect-mri-research-collaboration/gemric-the-global-ectmri-research-collaboration.

## Additional information

### Competing interests

Marom Bikson: reports that The City University of New York (CUNY) has intellectual property (IP) on neuro-stimulation system and methods with MB as inventor; serves on the scientific advisory boards of Boston Scientific and GSK; has equity in Soterix Medical. Anders M Dale: reports that he is a Founder of and holds equity in CorTechs Labs, Inc, and serves on its Scientific Advisory Board. He is a member of the Scientific Advisory Board of Human Longevity, Inc and receives funding through research agreements with General Electric Healthcare and Medtronic, Inc. The other authors declare that no competing interests exist.

### Funding

| Funder | Grant reference number | Author |
|---|---|---|
| National Institute of Mental Health | MH092301 | Randall Espinoza Katherine L Narr |
| National Institute of Mental Health | MH110008 | Randall Espinoza Katherine L Narr |
| National Institute of Mental Health | MH102743 | Randall Espinoza Katherine L Narr |
| Western Norway Regional Health Authority | 911986 | Ketil J Oedegaard |
| Western Norway Regional Health Authority | 912238 | Leif Oltedal |
| Deutsche Forschungsgemeinschaft | DFG FOR2107 | Udo Dannlowski |
| Deutsche Forschungsgemeinschaft | SFB-TRR58 | Udo Dannlowski |
| Deutsche Forschungsgemeinschaft | Project C09 | Udo Dannlowski |
| Innovative Medical Research | RE111604 | Ronny Redlich |
| National Institute of Mental Health | ZIAMH00295 | Zhi-De Deng |
| National Institute of Mental Health | MH119616 | Miklos Argyelan |
| National Institute of Mental Health | MH111826 | Christopher Abbott |
| Deutsche Forschungsgemeinschaft | DA1151/5-1 | Udo Dannlowski |
| Deutsche Forschungsgemeinschaft | DA1151/5-2 | Udo Dannlowski |
| Innovative Medical Research | RE111722 | Ronny Redlich |
| Deutsche Forschungsgemeinschaft | Project Z02 | Udo Dannlowski |
| University of Bergen | Fulbright Program | Leif Oltedal |

The funders had no role in study design, data collection and interpretation, or the decision to submit the work for publication.

## Author contributions

Miklos Argyelan, Conceptualization, Resources, Data curation, Software, Formal analysis, Supervision, Validation, Investigation, Visualization, Methodology, Writing—original draft, Project administration, Writing—review and editing; Leif Oltedal, Conceptualization, Resources, Data curation, Software, Formal analysis, Supervision, Funding acquisition, Validation, Visualization, Methodology, Writing—original draft, Project administration, Writing—review and editing; Zhi-De Deng, Supervision, Methodology, Writing—review and editing; Benjamin Wade, Conceptualization, Software, Formal analysis, Methodology, Writing—original draft, Writing—review and editing; Marom Bikson, Conceptualization, Software, Methodology, Writing—review and editing; Andrea Joanlanne, Data curation, Writing—review and editing; Sohag Sanghani, Marta Cano, Akihiro Takamiya, Jesus Pujol, Conceptualization, Writing—review and editing; Hauke Bartsch, Resources, Software, Formal analysis, Methodology, Project administration; Anders M Dale, Conceptualization, Resources, Data curation, Software, Methodology; Udo Dannlowski, Annemiek Dols, Conceptualization, Data curation, Project administration, Writing—review and editing; Verena Enneking, Randall Espinoza, Ute Kessler, Ketil J Oedegaard, Mardien L Oudega, Ronny Redlich, Max L Stek, Louise Emsell, Filip Bouckaert, Pascal Sienaert, Conceptualization, Data curation, Writing—review and editing; Katherine L Narr, Conceptualization, Data curation, Supervision, Methodology, Writing—review and editing; Indira Tendolkar, Conceptualization, Supervision, Project administration, Writing—review and editing; Philip van Eijndhoven, Conceptualization, Data curation, Methodology, Writing—review and editing; Georgios Petrides, Conceptualization, Resources, Data curation, Methodology, Writing—original draft, Project administration, Writing—review and editing; Anil K Malhotra, Conceptualization, Resources, Data curation, Methodology, Writing—review and editing; Christopher Abbott, Conceptualization, Resources, Data curation, Software, Formal analysis, Supervision, Investigation, Visualization, Methodology, Writing—original draft, Project administration, Writing—review and editing

## Author ORCIDs

Miklos Argyelan (iD) https://orcid.org/0000-0002-7254-1776
Marta Cano (iD) http://orcid.org/0000-0003-0675-9483

## Ethics

Human subjects: All sites' contributing data received approval by their local ethical committees or institutional review board, and the centralized mega-analysis was approved by the Regional Ethics Committee South-East in Norway (2013/1032 ECT and Neuroradiology, June 1, 2015).

## Decision letter and Author response

Decision letter https://doi.org/10.7554/eLife.49115.SA1
Author response https://doi.org/10.7554/eLife.49115.SA2

## Additional files

### Supplementary files

• Source code 1. Once EF and ΔVol were collected across 85 ROI in 151 subjects, all the rest of the calculations were carried out in R environment.This file is the source code of these calculations organized in org-mode (https://orgmode.org/) for reproducibility.

• Supplementary file 1. Volume changes following electroconvulsive treatment (ECT).One sample t-test in each ROIs. The table indicates t, uncorrected p, Cohen's d effect size, and FDR corrected p values. This sample is a sub-cohort of a recent publication of 331 subjects (*Ousdal et al., 2019*). In contrast with that publication we included patients with RUL only electrode placement.

• Supplementary file 2. The laterality of volume changes after RUL ECT.The table contains the results of 42 pairwise t-tests between the volume changes of the right and the left side of the

corresponding regions. The table indicates t, uncorrected p, mean, Cohen's d effect size and FDR corrected p values.

• Supplementary file 3. The relationship between clinical response and volume change across individuals.The table indicates the t values of the corresponding clinical covariates modeled as fixed effects: volume change, age, and number of ECT.

• Supplementary file 4. The relationship between clinical response and electric field across individuals.The table indicates the t values of the corresponding clinical covariates modeled as fixed effects: electric field, age, and number of ECT.

• Supplementary file 5. The relationship between the baseline volume and age across 85 ROIs.

• Supplementary file 6. MRI summary.The table indicates the parameters of the structural image acquisition across sites.

• Transparent reporting form

### Data availability

Source data files (.csv) including processed data have been provided for Figures 1–4, Figure 3—figure supplement 1 and Figure 4—figure supplement 1. Raw data cannot be made available publicly because we do not have consent or ethical approval for this and the data cannot be anonymised. The data are stored on a secure centralized server at the University of Bergen, Norway. Participating GEMRIC sites have access to the raw data according to specific data policy and safety rules of the consortium, and in accord with the approval from the ethical committee. The GEMRIC consortium welcomes new members who are interested in the neuroimaging research of ECT. We hold board meetings twice a year when new members can apply to join and gain access to the database available on the GEMRIC servers. For more about the application process please visit https://mmiv.no/how-to-join-gemric/ or write to Leif Oltedal (leif.oltedal@uib.no). General information about the consortium can be found on the following website: https://mmiv.no/gemric/. For transparency and reproducibility, the entire analytical approach is uploaded to the https://github.com/argyelan/Publications/tree/master/ECTEFvsVOLUME (R scripts in org mode), and also was uploaded to eLife.

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
