## [Decision Letter]

**Acceptance summary:**

This analysis of data from the GEMRICS consortium on electroconvulsive therapy (ECT) investigates the potential links between electric field, brain volume change, and antidepressant response. Using data from 151 patients, the study finds a significant volume increase in the large majority of brain regions examined and a significant relation between electric field and volume change, particularly in left hippocampus and left amygdala. By contrast, it fails to find a significant relation between either electric field or volume change and clinical response. Given the importance of ECT as a last resort for treating severe depression and the necessity of better understanding its mechanisms, this study contributes important biological findings that will guide future investigations on the mechanisms of ECT.

**Decision letter after peer review:**

Thank you for submitting your article "Electric field causes volumetric changes in the human brain" for consideration by *eLife*. Your article has been reviewed by two peer reviewers, including Klaas Enno Stephan as the Reviewing Editor and Reviewer #1, and the evaluation has been overseen by Christian Büchel as the Senior Editor. The following individuals involved in review of your submission have agreed to reveal their identity: Juergen Dukart (Reviewer #2).

The reviewers have discussed the reviews with one another and the Reviewing Editor has drafted this decision to help you prepare a revised submission.

Summary:

This paper reports an analysis of data from the GEMRICS consortium on electroconvulsive therapy (ECT), investigating the potential links between electric field, brain volume change, and antidepressant response. Using data from 151 patients, the study finds a significant volume increase in 82 of 85 brain regions examined and a significant relation between electric field and volume change in left hippocampus and left amygdala. By contrast, it fails to find a significant relation between either electric field or volume change and clinical response.

Given the importance of ECT as a last resort for treating severe depression and the necessity of better understanding its mechanisms, this study addresses a highly relevant and timely issue. Both reviewers were generally positive about this paper. However, they also identified several issues that would require revisions and additional analyses.

Essential revisions:

1) It did not get entirely clear to me whether overlapping questions (or perhaps even the same) have been asked in previous analyses using data from the GEMRICS consortium. I do not necessarily consider this a problem, as long as the data are different, but it should be transparent for the reader.

2) Results section: the descriptions of the findings in Figure 4 (i.e., the alleged ceiling effect in the right hemisphere) are rather cryptic in the main text and deserve more explanation. I also wondered whether the difference between volume changes in left and right hemisphere (the right panel of Figure 4) is actually significant? Also, it would be helpful if you could comment on how this putative ceiling effect is compatible with the findings presented in Figure 3.

3) The central findings reported in the subsection "Electric field, volume change, and clinical response" would deserve a considerably more detailed description. There are no space constraints in *eLife*, and it would be helpful to see a detailed breakdown of the statistical analyses in the main text (not just in the supplementary material). In particular, it would be nice to see the relative influence of each component of the regression model. For example, one would like to see in the main text how strong the effect of age was on clinical response to ECT.

4) The authors used% changes in MADRS as the outcome. This choice is problematic as a reduction from 2 to 1 point is a 50% reduction whilst a reduction from 40 to 30 is only 25%. This choice may therefore create quite a biased evaluation of clinical improvement. The authors are therefore advised to repeat the analyses using the absolute scores as more unbiased measurement of clinical response.

5) In the model evaluating the effects of volume change on MADRS the authors also include number of ECT sessions as a covariate. This is problematic from a conceptual point of view. The authors aim to study the mediating effect of volume changes induced by ECT on clinical symptoms. Controlling for the number of ECT sessions removes exactly this variance of interest from volume changes. The authors should therefore repeat the analyses without adjusting for this covariate, or perhaps even more convincingly, perform a mediation analysis that disentangles the direct and indirect (via volume changes) effects of number of ECT sessions on MADRS scores. Similar concern applies to age as a mediator of the effect of ECT.

6) Please add a more detailed description of the overall cohort (i.e. sex, medication status.…) to Table 1A, also not only per site but for the whole cohort and move it to the main body as it is important for a clinical study to have this information in the main body.

7) At least some studies suggest potentially differential relationship between volume changes upon ECT and clinical response in bipolar and unipolar depression. Whilst the bipolar group may be not large enough it makes sense to test for this relationship separately for both diagnostic groups, or at least for major depression.

8) Lastly, I am concerned by the highly significant volume increases in basically all brain regions in both hemispheres reported in Supplementary file 2 (excluding only the cerebellum). These effects go basically against all previous studies reporting rather quite specific structural changes upon ECT. The authors seem to use some very specific and a rather uncommon software package to compute these changes which raises the question of a methodological issue with the respective software. As the authors already have the free surfer estimates, I would ask them to compute directly the volume changes from those just to confirm that the effects are not dependent on the respective software choice.

[Editors' note: further revisions were requested prior to acceptance, as described below.]

Thank you for resubmitting your work entitled "Electric field causes volumetric changes in the human brain" for further consideration at *eLife*. Your revised article has been favorably evaluated by Christian Büchel (Senior Editor) and Klaas Enno Stephan (Reviewing Editor).

The manuscript has been improved but there are some remaining issues that need to be addressed before acceptance, as outlined below:

Point 4: In your response, you referred to a "complementary analysis with baseline MADRS score as a covariate" which "resulted in similar findings to our% MADRS." We may have overlooked this but failed to find this analysis. We would like to ask that this complimentary analysis is included in the main text (or in the supplementary material, with a direct reference in the main text) so that reviewers and readers can see the similarities directly.

Point 5: Given that this is a question many readers may wonder about, it would be worth to include your detailed explanations (why no mediation analysis was performed and why "number of ECT sessions" was included as a covariate in the statistical model) in the Discussion section of the paper.

Point 8: We would suggest that the previous validation of the analysis pipeline against other software packages is described in the Discussion section of the paper, together with your interpretation why previous studies failed to find widespread volume changes, in order to make this issue transparent.

---

## [Author Response]

Essential revisions:1) It did not get entirely clear to me whether overlapping questions (or perhaps even the same) have been asked in previous analyses using data from the GEMRICS consortium. I do not necessarily consider this a problem, as long as the data are different, but it should be transparent for the reader.

The EF modeling data has not been included in previous GEMRIC analyses; the analysis of EF modeling with volume change (the main finding) and clinical response is completely new. It is correct that the broad distribution of volume change and relation to clinical response was described in more detail in a separate recent GEMRIC paper (Ousdal et al., 2019). However, that analysis also included patients stimulated with bilateral electrode placements. The present investigation includes only subjects treated with right unilateral electrode placement and e-field analysis. We have clarified the differences with the current work from previous work as follows in the concluding sentence of our Introduction: “For the purpose of our primary research question and in contrast to previous GEMRIC investigations, we limited the analyses to subjects that only received right unilateral electrode placement.”

2) Results section: the descriptions of the findings in Figure 4 (i.e., the alleged ceiling effect in the right hemisphere) are rather cryptic in the main text and deserve more explanation. I also wondered whether the difference between volume changes in left and right hemisphere (the right panel of Figure 4) is actually significant? Also, it would be helpful if you could comment on how this putative ceiling effect is compatible with the findings presented in Figure 3.

We have added to our Discussion section to further explain the non-linear relationship between EF and ∆Vol as follows: “Finally, the volume change required for response may be non-linear. A minimum electric field of 30-40 V/m may be necessary to induce neuroplasticity. Increasing the electric field between 30-40 V/m and 100 V/m is related to a monotonic increase in hippocampal volume. Electric field above 100 V/m is still associated with hippocampal neuroplasticity but the dose-response relationship may be less robust and represent a ceiling effect of electric-field induced neuroplasticity as illustrated in Figure 4. Surpassing the neurpolasticity threshold (100 V/m) appears to be unrelated to further volumetric increases and antidepressant response. Thus, the relationship between e-field and volumetric changes may be conceptualized as a “neuroplasticity threshold” between 30-40 V/m and 100 V/m. This thresholding effect also preserves the laterality of electric field and neuroplasticity. Our sample was limited to right unilatleral electrode placement. In the left hippocampus, the maximum electric field is ~ 80 V/m (Figure 3) and below the 100 V/m “ceiling effect” noted in the right hemisphere. Consequently, in our right unilateral sample, hippocampal electric field and related changes in neuroplasticity will demonstrated laterality effects.”

The difference in right and left hippocampal volume changes (right panel of Figure 4) is signficant (t = 7.76, df = 150, mean difference = 0.011, p < 0.0001). We have changed the Figure text for more clarity.

Figure 3 demonstrates laterality differences in EF and ∆Vol (upper panel) as well as the relationship between laterality between EF/∆Vol (lower panel). We have added to our Discussion section to furhter describe this relationship as noted above.

3) The central findings reported in the subsection "Electric field, volume change, and clinical response" would deserve a considerably more detailed description. There are no space constraints in eLife, and it would be helpful to see a detailed breakdown of the statistical analyses in the main text (not just in the supplementary material). In particular, it would be nice to see the relative influence of each component of the regression model. For example, one would like to see in the main text how strong the effect of age was on clinical response to ECT.

We agree with this suggestion, so we made the following changes:

a) We moved Supplementary file 4, which is our main finding, to Table 1, making the relative influence of the covariates more transparent. Also we extended the previous paragraph as follows: “Age was a necessary covariate since it was a confound in our model: both the spatial distribution of EF and volume changes correlate with age (30). We add number of ECT as a covariate to the model to be able to compare the relative influence of EF and number of ECT on volume change. In both left hippocampus and amygdala the effect size of EF was the largest (hippocampus: t_EF_ = 4.5, t_Age_ = -2.7, t_ECTnum_ = 3.3, amygdala: t_EF_ = 3.9, t_Age_ = -1.1, t_ECTnum_ = 2.1; Table 1).”

b) We also extended subsection “Electric field, volume change, and clinical response” with more details: “Results indicated that none of the volume changes across the 85 ROIs had a significant relationship with clinical response (Supplementary file 4, hippocampus: t_ΔVOL_ = 0.2, t_Age_ = 5.4, t_ECTnum_ = -2.7, amygdala: t_ΔVOL_ = 0.1, t_Age_ = 5.6, t_ECTnum_ = -3.0). These results therefore contradicted the hypothesis that EF by increasing brain volume indirectly exerts its effect on clinical response, given the negative results between the volume change (mediator) and MADRS change (outcome). Testing the direct effect of the EF, we failed to find significant correlations between EF and clinical response (Supplementary Table 5, hippocampus: t_EF_ = 1.2, t_Age_ = 5.7, t_ECTnum_ = -3.0, amygdala: t_EF_ = 1.1, t_Age_ = 5.7, t_ECTnum_ = -3.0). Similar to earlier studies, age strongly correlated with both clinical response (32, 33, also see Clinical Results) and EF distribution (30), therefore we controlled for age in our model. The rationale for including the number of ECT treatments as covariate needs more explanation. Due to the naturalistic nature of the design, where most sites followed the patient until response or site-determined criteria for ECT discontinuation, we observed a negative relationship between clinical response and the number of ECT treatments. Not controlling for this variable could lead to spurious correlation between volume change and clinical response (for more on this see Oltedal et al., 2018 (3)).”

c) For more detailed rationale for controlling these covariates please see our discussion under point 5.

d) We would like to also note that in the beginning of the Results section we have already explicitly stated the relationship between age and clinical response: “Highly significant correlations between age and clinical response (t = 5.75, df = 149, r = 0.43, p < 10^-7^, older patients responded better).”

4) The authors used% changes in MADRS as the outcome. This choice is problematic as a reduction from 2 to 1 point is a 50% reduction whilst a reduction from 40 to 30 is only 25%. This choice may therefore create quite a biased evaluation of clinical improvement. The authors are therefore advised to repeat the analyses using the absolute scores as more unbiased measurement of clinical response.

Based on these suggestions we used absolute scores in a complementary analsyis with baseline MADRS score as a covariate to control for baseline severity, and it resulted in similar findings to our% MADRS.

All subjects had a severe depressive episode (mean pre-ECT MADRS ~ 33.9). The rationale for modeling proportional change instead of absolute change score or outcome severity directly was to control for the pre-ECT MADRS variability. We added the following test (and reference) to our methods section: “Although more conservative than absolute change or post-ECT depression outcomes (Vickers, 2001), the rationale for the use of the proportional change score was to control for the variability of the pre-ECT MADRS.”

5) In the model evaluating the effects of volume change on MADRS the authors also include number of ECT sessions as a covariate. This is problematic from a conceptual point of view. The authors aim to study the mediating effect of volume changes induced by ECT on clinical symptoms. Controlling for the number of ECT sessions removes exactly this variance of interest from volume changes. The authors should therefore repeat the analyses without adjusting for this covariate, or perhaps even more convincingly, perform a mediation analysis that disentangles the direct and indirect (via volume changes) effects of number of ECT sessions on MADRS scores. Similar concern applies to age as a mediator of the effect of ECT.

We agree that mediation analysis would be the most interesting part of this dataset as it could support mechanistic understanding. Mediation analysis in general means that the following three conditions are met: (1) there is a significant correlation between outcome and mediator, (2) there is a significant correlation between predictor and outcome, and (3) in a multiple regression model where we model outcome~mediator + predictor the mediator’s coefficient remain significant, and at the same time the predictor’s coefficient is not significant anymore. The reviewer is correct that our original goal was to show that volume change is the mediator of the clinical effect. Our original hypothesis was that EF -> volume change -> clinical response. (-> means: cause) However, since volume change has no correlation with clinical change, neither in this dataset, nor in the recently published larger set of data with more heterogeneous ECT electrode placement (1), the first condition of mediation analysis is already invalid. Therefore, formal mediation analysis (i.e. Sobel test) is not appropriate for this dataset. In regard of the number of treatments we agree with the reviewer that careful consideration is required in adding the number of ECT treatment variable (see rationale under point 3). Here we provide extra details. Due to the naturalistic design, where most sites followed the patient until response or site-determined criteria for ECT discontinuation, “Number of ECT” could either be a mediator (Author response image 1, Model 1) or a confound (Author response image 1, Model 2). Regardless of the role of “Number of ECT” as mediator or confound, we believe that inclusion in our model is necessary because of the naturalistic, multi-site design of our dataset. The remaining correlation can reflect the direct relationship between volume change and MADRS change.

In our earlier paper (Oltedal et al., 2018) we already encountered this problem because of this relationship between number of ECT and clinical response. Originally, we found a mild effect between hippocampus change and clinical response, but, contra-intuitively, number of ECT treatments reflected increased volume change that was associated with worse outcome (Oltedal et al., 2018). However, this relationship was completely absent when we controlled for the number of ECTs. This can be demonstrated via mediation analysis similar to which the reviewer proposed. Mediation analysis in our sample resulted in p = 0.035 and p = 0.034 in L and R Hippocampus reflectively (Sobel test). These results indicate that the relationship between clinical response and volume change in these areas are fully mediated via number of ECT sessions, therefore we should control for it in subsequent analyses.

6) Please add a more detailed description of the overall cohort (i.e. sex, medication status.…) to Table 1A, also not only per site but for the whole cohort and move it to the main body as it is important for a clinical study to have this information in the main body.

We added a new Table to the main text that contains the required clinical and demographic information.

7) At least some studies suggest potentially differential relationship between volume changes upon ECT and clinical response in bipolar and unipolar depression. Whilst the bipolar group may be not large enough it makes sense to test for this relationship separately for both diagnostic groups or at least for major depression.

We have tested this hypothesis, however it did not change the results if we excluded the 12 bipolar patients from our sample. We inserted the following test in the manuscript: “The results did not change if we used medication status or diagnosis (bipolar or unipolar depression) as nuisance variables in the linear models of this study.”

8) Lastly, I am concerned by the highly significant volume increases in basically all brain regions in both hemispheres reported in Supplementary file 2 (excluding only the cerebellum). These effects go basically against all previous studies reporting rather quite specific structural changes upon ECT. The authors seem to use some very specific and a rather uncommon software package to compute these changes which raises the question of a methodological issue with the respective software. As the authors already have the free surfer estimates, I would ask them to compute directly the volume changes from those just to confirm that the effects are not dependent on the respective software choice.

The processing pipeline that was used has been validated against many commonly used tools for estimating longitudinal volume change (Holland and Dale, 2011; Holland, McEvoy and Dale, 2012. Specifically, it was previously compared head-to-head with FreeSurfer 5.3, and we have already repeated this comparison for data from one of the GEMRIC sites (Oltedal et al., 2017). Our comparisons of power estimations based on results from the FreeSurfer longitudinal pipeline and Quarc (Table 3 in Oltedal et al. 2017) were in line with those of the earlier publications. The broad distribution of volume change is described in more detail in a separate recent GEMRIC paper (Ousdal et al., 2001) and was already noted in terms of effect sizes in Figure 3 in Oltedal et al., 2017, see response to question 1. In agreement with previous research, the effect sizes show regional differences indicating that earlier studies with lower sample sizes were underpowered to detect cortical changes, and that can explain why they only found subcortical volume increase.

Furthermore, using the same methodology, we did not find any significant volume change in the 95 healthy controls (received no ECT) who were imaged at two time points (Osudal et al., Supplementary Table 4).

[Editors' note: further revisions were requested prior to acceptance, as described below.]

The manuscript has been improved but there are some remaining issues that need to be addressed before acceptance, as outlined below:Point 4: In your response, you referred to a "complementary analysis with baseline MADRS score as a covariate" which "resulted in similar findings to our% MADRS.” We may have overlooked this but failed to find this analysis. We would like to ask that this complimentary analysis is included in the main text (or in the supplementary material, with a direct reference in the main text) so that reviewers and readers can see the similarities directly.

We have included these results in the second half of the Supplementary file 3 and Supplementary file 4. It is now indicated in the Materials and methods section: “In addition to analyzing the percentage change of the clinical response, we also evaluated the same models with absolute changes, using baseline MADRS as a covariate. We provided the results of these analyses in the second half of the corresponding Supplementary files.”

Point 5: Given that this is a question many readers may wonder about, it would be worth to include your detailed explanations (why no mediation analysis was performed and why "number of ECT sessions" was included as a covariate in the statistical model) in the Discussion section of the paper.

We have included two paragraphs in the Discussion section:

“Our original hypothesis was that a) local electric field had a causal role in clinical outcome and that b) the corresponding volume change was mediating this relationship […] However, since volume change showed no correlation with clinical change, neither in this dataset, nor in the recently published broader dataset with more heterogeneous ECT electrode placement (10), only the first half of this model, namely that electric field strength was associated with volume change, was supported by our data.”

“Additionally, it was necessary to control in our regression models for the number of ECT treatments. […] Mediation analysis supported a very similar situation in our sample with p = 0.035 and p = 0.034 in L and R Hippocampus reflectively (Sobel test).

It was, therefore, necessary to control for the number of ECT sessions to avoid detecting spurious correlations between clinical response and volume change.”

Point 8: We would suggest that the previous validation of the analysis pipeline against other software packages is described in the Discussion section of the paper, together with your interpretation why previous studies failed to find widespread volume changes, in order to make this issue transparent.

We have included the following paragraph:

*“*Our findings indicate widespread and robust volume changes in both cortical and subcortical regions. […] Furthermore, using the same methodology, we did not find any significant volume change in the 95 healthy controls (received no ECT) who were imaged at two time points (10)*”.*